# A Stereoselective Synthesis of a Novel α,β-Unsaturated Imine-Benzodiazepine through Condensation Reaction, Crystal Structure, and DFT Calculations

**DOI:** 10.3390/molecules29184323

**Published:** 2024-09-12

**Authors:** Samir Hmaimou, Marouane Ait Lahcen, Mohamed Adardour, Mohammed M. Alanazi, Atul Kabra, Mohamed Maatallah, Abdesselam Baouid

**Affiliations:** 1Molecular Chemistry Laboratory, Department of Chemistry, Semlalia Faculty of Sciences, Cadi Ayyad University, Marrakech 40001, Morocco; samir.hmaimo@gmail.com (S.H.); mar.aitlahcen@gmail.com (M.A.L.); adardourmed@gmail.com (M.A.); baouid@uca.ac.ma (A.B.); 2Department of Pharmaceutical Chemistry, College of Pharmacy, King Saud University, Riyadh 11451, Saudi Arabia; mmalanazi@ksu.edu.sa; 3University Institute of Pharma Sciences, Chandigarh University, Mohali 14041, Punjab, India; atul.e9963@cumail.in

**Keywords:** benzodiazepine, condensation, stereoselective, X-ray, DFT calculation

## Abstract

The stereoisomers (E)-2,2-dimethyl-4-(4-subsitutedstyryl)-2,3-dihydro-1H-[1,5]-benzodiazepine **3(a–d)** were synthesized via the condensation reaction of 2,2,4-trimethyl-2,3-dihydro-1H-1,5-benzodiazepine (BZD) **1** with the benzaldehyde derivatives **2(a–d)** in ethanol. The chemical structure of the prepared products was confirmed by NMR (^1^H and ^13^C), HRMS, and X-ray analysis of the crystal structure **3d**. The condensation reaction was examined using DFT calculations at the theoretical level of B3LYP/6-31G(d) to elucidate the chemo-, regio-, and stereoselectivity and the reaction mechanism of the produced isomer. Furthermore, we identified each reagent’s reactive sites by the measurement of the reactivity indices. We also looked at how the electron-withdrawing groups (EWGs) of various aldehydes affected the reaction’s mechanism and the stability of products **3(a–d)**.

## 1. Introduction

In recent years, the importance of bioactive heterocycle compounds especially benzodiazepines (BZDs) has been underlined [1,2,3,4]. Benzodiazepines have received great interest due to their reactivity and diverse biological activities, including hypnotic sedatives and anxiolytic, antifungal, antibacterial, anti-tubercular, and cytotoxicity effects [5,6,7,8,9]. In addition to their traditional use in biology and medicinal chemistry, benzodiazepines have demonstrated potential as effective inhibitors in many investigations [10,11,12]. Since the introduction of heteroatoms and multiple bonds greatly increases the inhibition performance of organic compounds, chemists are interested on developing new methods to incorporate these elements via condensation reactions, substitutions, and cycloadditions [13,14,15].

Today, many routes for the synthesis of BZDs have been described, employing classical and catalytic methods [16,17]. Recently, the synthesis of new interesting heterocyclic compounds in the organic and biological field, with fewer steps, has been challenging for organic chemists [18]. However, various reaction protocols using both synthetic methods that allow access to diverse functionalization of benzodiazepines have been established for efficient and higher yields [19].

Generally, BZDs are a key intermediate structure for preparing the other BZD derivatives with a new reactive center [20] and for preparing novel pentagonal molecules via cycloaddition reaction [21,22,23]. On the other hand, the most facile method for the synthesis of α,β-unsaturated imine is the reaction of condensation of α,β-unsaturated ketones with amines [24]. Also, aldehydes are known to react with α-hydrogen of imine to synthesize α,β-unsaturated imines, which can also be employed as precursors for the preparation of new pharmacologically active heterocycles [20].

Many quantum methods are highly helpful for comprehensively studying condensation reactions [24], with particular emphasis on elucidating their chemo- and regioselectivities and their mechanism [25]. The molecular mechanism, substrate reactivity, and factors controlling chemo-, regio-, and stereoselectivity involved in this type of organic reaction are definitively established using the density functional theory (DFT) [26,27]. In addition, in-depth studies of the reaction mechanism provide a clear elucidation of the influence of electronic and steric effects on both the reaction and its reactivity [25,28,29,30].

As part of our ongoing interest in the efficient synthesis of new 1,5-benzodiazepine derivatives, we herein report the unique strategy for the one-pot synthesis of α,β-unsaturated imines via condensation reaction of our 1,5-benzodiazepines (BZDs) **1** with benzaldehyde derivatives **2(a–d)**. All the structures of prepared compounds were determined by NMR (^1^H and ^13^C) and HRMS and confirmed by the results of the crystal analysis of compound **3d**. This condensation process’s chemo- and stereoselectivity have been investigated experimentally and theoretically using the molecular electronic system density theory. The results of the Fukui function analysis of the reactants’ electronic localization function accurately anticipated the experimental outcome. In addition, we approved the mechanism of the condensation reaction, and we studied the effect of the electron-withdrawing group (EWG) of the R substituent of aldehydes **2(a–d)** on the studied reaction.

## 2. Results and Discussion

### Synthesis

The target compounds **3**(**a–b**) were synthesized by condensation of BZD **1** with benzaldehyde derivatives. In the first step, BZD **1** was synthesized using the literature approach [31]. Then, this BZD reacts with benzaldehydes **2**(**a–d**) analogues in ethanol to reflux; the reaction gives the desired products with good yield (Figure 1). It should be noted that compound **3a** has already been synthesized by Odame et al. [20], and for this study, we will take it as a reference to compare our results in terms of target product geometry, reaction time, yield, and reactivity.

The reactions were carried out in ethanol, serving as a solvent and a catalyst. In theory, four synthesis pathways are feasible for our reaction: two reactions result in a condensation reaction to give products **3(a–d)** and **4(a–d)** with the elimination of a water molecule (Path A and Path B), while the other two reactions result in an addition reaction on the nitrogen atoms to obtain products **5(a–d)** and **6(a–d)** (Path C and Path D). Due to the strict hindrance of the methyl groups, the addition reactions are difficult [22], whereas the condensation reaction is particularly possible since there is no strict hindrance.

These condensation reactions began with the activation of carbon in the position α of the imine group by the proton of the ethanol and then its attack of the carbonyl group of benzaldehyde to give theoretically two isomers, isomer (E) **3(a–d)** and isomer (Z) **4(a–d)**, with the elimination of a water molecule. Experimentally, just isomer (E) was observed due to its stability, and their structures were confirmed by ^1^H, ^13^C NMR, and X-ray analysis of **3d**. These results explain that the reaction is completely regio- and stereoselective.

The ^1^H NMR spectra of our products show two overlapping peaks at about 1.23 ppm due to the CH_3_ protons of the molecule, a singlet around 2.42 ppm due to the CH_2_ protons in position 2 of the diazepine ring, and the other signals refer to the two CH protons of the structure. Their ^13^C NMR spectra also illustrate two peaks at 30 ppm, attributed to the two primary carbons of the molecules; two other peaks at 40.27 and 68.03 ppm attributed to the secondary and quaternary carbon, respectively; and the main peak close to 168 ppm attributed to the quaternary carbon C = N of the molecule (Appendix A).

## 3. Crystal Structure of Compound 3d

Figure 1 shows the crystal structure of compound **3d**. This compound crystallizes in an orthorhombic system with the space group Pna2_1_. The parameters of product **3d** are reported in Table 1. The value of Z’ is 2, which signifies that the asymmetric unit consists of two separate molecules. In addition, the molecules show weak H-bonding between the N_6_H and N_2_ atoms of nearby molecules, resulting in a polymeric strand in the solid state (Figure 2).

On the other hand, the flack parameters are used to identify the chirality of the crystal under study; a value of 1 indicates that the stereochemistry is wrong, and the model should be inverted. A value of 0.5 indicates that the crystal is composed of a racemic mixture of the two enantiomers. The flack parameter was set to 0.5, indicating that our molecule is a racemic mixture of the two enantiomers.

## 4. Theoretical Study

Understanding the chemo- and regioselectivity in organic reactions poses a significant challenge for experimenters. The study of organic compounds’ electronic, structural, and thermodynamic properties, as well as the understanding of the reaction mechanisms leading to a specific product, often requires complementary approaches to experimentation alone. Quantum computational methods, particularly those using the density functional theory (DFT), have been invaluable in studying and understanding the reactivity of organic compounds and their thermodynamic and kinetic properties [32,33,34].

As part of this study, we performed quantum calculations based on the DFT using the B3LYP function. The effect of the solvent on the reactivity of the compounds was examined by the conductive-like polarizable continuum model (CPCM). First of all, we carried out a thermodynamic analysis to identify the most stable products resulting from this reaction. Then, we studied the chemical reactivity of these molecules by exploring the global chemical descriptors, which allowed us to identify the character of our reagents (nucleophile or electrophile species). Finally, we analyzed the energy profile of the reactions leading to the formation of compounds **3(a–d)**.

Theoretically, the reaction between BZD **1** and different aldehydes **2(a–d)** can produce several products. Figure 1 shows the main products of this reaction, and Table 2 lists the total energy (E) and relative energy (DE), which represents the energy difference between the most stable regioisomer and the other. From the thermodynamic point of view, the results of the analysis of these energy values reveal that products **3(a–d)** and **4 (a–d)** are more stable, and, therefore, they are more likely to form in our reaction. On the other hand, products **5(a–d)** and **6(a–d)** have high energies and are, therefore, thermodynamically disadvantaged.

The stability of **3(a–d)** and **4(a–d)** products in comparison with products **5(a–d)** and **6(a–d)** is explained by the formation of a new double bond (C_8_=C_7_) established between the carbonyl group of the aldehyde and the C_8_ site of BZD **1**. By taking part in conjugation, this bond increases the molecule’s stability. This stability is further supported by the absence of a steric gene at this location (C_8_ of BZD **1**). On the other hand, the instability of compounds **5(a–d)** and **6(a–d)** is due to the presence of the steric gene of the methyl groups of BZD **1**. In addition, the instability of the bond that can form between BZD **1** and benzaldehyde **2(a–d)** (N_2_-C_7_ or N_3_-C_7_) is explained by the fact that the C_7′_ carbon is surrounded by heteroatoms (N and O) and a phenyl group, which are groups that attract electrons through inductive effects. In chemical terms, this carbon becomes unstable, which makes the chemical reaction difficult to take place in this position. These results confirm once again that the reaction is completely chemoselective in which products **3(a–d)** represent that trans isomer is more stable than cis isomer products **4(a–d)**.

### Analysis of the Conceptual DFT Reactivity Indices of the Reagents

In the molecule, the highest occupied molecular orbital (HOMO) is an electron-rich orbital, acting as an electron donor during a chemical reaction. On the other hand, the lowest unoccupied molecular orbital (LUMO) is an electron-poor orbital, responsible for accepting electrons [35]. These two molecular orbitals are used to calculate the indices of chemical reactivity, which makes it possible to understand the stability and reactivity of a molecular system, as well as its electronic behavior in a chemical reaction [36].

The main chemical reactivity indices, determined from the HOMO and LUMO frontier orbitals and their corresponding energies (E_HOMO_ and E_LUMO_ respectively), include the ionization potential (I) related to the energy of the molecule, and measuring the energy necessary to yield an electron; and the affinity (A) related to the LUMO energy, which indicates the energy required for the molecule to accept an electron in the LUMO. The chemical hardness (η) and the chemical softness (σ) of the molecule are both significantly related to the energy gap (ΔE) between the HOMO and the LUMO. The chemical hardness (η) of a molecule is determined by using the formula ƞ = ΔE/2 to indicate its resistance to the transfer of charge carriers. On the other hand, its softness (σ) is calculated by using the formula σ = 1/2 ƞ, which indicates its capacity to hold into an acquired charge. The electronegativity (χ) of the molecule, which is related to its propensity to attract electrons, is expressed by χ = (I + A)/2. The chemical potential (μ), which quantifies a molecule’s capacity to transmit electrons, is expressed in μ = −χ. A molecule’s global electrophilicity (ω = μ^2^/2̞) indicates its ability to acquire an electronic charge, while its global nucleophilicity (N), which indicates its ability to react as a nucleophile, can be calculated using the formula E_HOMO_ − E_HOMO_(TCE), where E_HOMO(TCE)_ = −9.3686 eV [37,38].

We calculated their global reactivity indices to evaluate the acceptor or electron donor character of our reagents. The energy values of the main index are presented in Table 3. A more favored and more possible reaction is suggested by a small energy gap (ΔE) between the HOMO and the LUMO of the reactants, as this facilitates the exchange of electrons between the HOMO and the LUMO. The calculated energy differences (ΔE1) between the HOMO of BZD1 and the LUMO of **2(a–d)**, as shown in Figure 3, are 3.668, 3.278, 2.149, and 3.553 eV, respectively; while the energy differences (ΔE2) between the HOMO of **2(a–d)** and the LUMO of BZD 1 are 6.294, 6.603, 7.026, and 6.396 eV, respectively. These values are higher than those of ΔE1, which suggests a typical condensation reaction, where BZD1 acts as a nucleophile with a high HOMO energy. In contrast, **2(a–d)** acts as an electrophile with a higher LUMO energy.

The calculated reactivity indices confirm this result. Indeed, BZD **1** has a higher chemical potential (4720 eV) compared with benzaldehydes **2(a–d)** (−2.227, −2.580, −2.621, and −2.614 eV, respectively), which suggests a transfer of electrons from BZD **1** to benzaldehydes **2(a–d)**. In addition, BZD **1** is recognized as a nucleophilic chemical species, with a global nucleophilicity of 4.103 eV and a global electronegativity of −2.905 eV. At the same time, benzaldehydes **2(a–d)** are electrophilic chemical species, with a higher global electrophilicity (−0.556, −0.645, −0.655, and −0.653 eV, respectively) and a low global electronegativity of 2.614, 2.621, 2.580, and 2.227 eV, respectively.

Electronic and steric effects are well identified as two major factors influencing the energy of the HOMO and the LUMO. The HOMO energy values for the para-NO_2_, para-Cl, para-CH_3_, and para-H substituents are −7.571, −7.148, −6.839, and −6.941, respectively; while their LUMO energy values are −3.116, −1.987, −1.597, and −1.712, respectively. The para-benzaldehyde substituent R (R = NO_2_ Cl, CH_3_, and H) impacts the energies of the HOMO and the LUMO. We observe that the NO_2_ and Cl attractor groups decrease the value of the HOMO while increasing that of the LUMO, and vice versa for the donor group (CH_3_). This suggests that the more the substituent is an attractor, the more the compound exhibits a significant electrophilic character, and vice versa. Thus, product **2d** is the most electrophilic with a lower softness value of −0.224 eV, followed by **2c** then **2a**, and finally **2b** is the least electrophilic with a higher softness value of −0.190 eV.

## 5. Kinetic Study

In this part, we examined the kinetics of the reaction of BZD **1** with aromatic aldehydes **2(a–d)** to better understand the mechanism and the stereochemistry of the products obtained. First of all, we optimized all the stationary points involved in the reaction mechanism at the B3LYP/6-31G(d) level. Then, we performed a frequency calculation to characterize the nature of these stationary points on the potential energy surface (PES); the minima are defined by a zero frequency and the transition states by an imaginary frequency. Intrinsic coordinates of the reaction (IRC) calculations were carried out to elucidate the different reaction paths. Figure 4 shows the optimized structures associated with the proposed reaction path, with the significant geometric parameters for each species involved in the reaction. The energy values relative to each stationary point, as a function of the nature of the substituent, are grouped in Table 4. The energy profile of the proposed reaction path is illustrated in Figure 5.

The proposed energy profile leads to the formation of two stereoisomers with the configuration (**Z**) and (**E**). The transition from BZD **1** to the final products involves two intermediates **In1** and **In2**, all located on the total PES. The reaction of BZD **1** with ethanol gives the intermediate **In1**, requiring the crossing of a transition state **TS1** with an activation energy of 27.092 kcal/mol (Figure 5). The reaction of the intermediate **In1** with the aldehyde **2(a–d)** leads to the formation of the second intermediate (**Int2**) present in two stereoisomers. This intermediate marks the formation of a new C_8_-C_aldehyde_ = O bond and results in the formation of an alcohol function in the form of two enantiomers of configuration (R) and (S). The two stereoisomers **In2 (R)** and **In2 (S)** are linked to the intermediate **In1** by a transition state **TS2**, confirmed by the IRC calculations, characterized by an activation barrier of 12.785, 13.439, 11.705, and 8696 Kcal/mol depending on the nature of the substituent R, which is equal to H, CH_3_, Cl, and NO_2_ of the aldehyde, respectively.

Theoretical calculations show that the stereoisomers (R) are thermodynamically more stable compared with the stereoisomers (S). This stability phenomenon can be explained by the conformation adopted by the stereoisomer formed as well as by the existence or absence of the steric gene of the two bulky groups linked to the functional carbon of the alcohol function (**In2**).

In the presence of ethanol, the intermediates **In2 (S)** and **(R)** lead, via the transition states **TS3(S)** and **TS3(R),** respectively, to the formation of two diastereomers of configurations (**E**) and (**Z**). The stereoisomer (**E**) is formed as a thermodynamic product of the reaction between ethanol and the intermediate **In2 (S)**, while (**Z**) results from the intermediate **In2 (R)**. Note that the activation energy necessary for forming the **E** stereoisomer is lower than that necessary to form the **Z**.

The mechanism of the reaction is explained by the passage of different transition states and intermediate leading to the final product **3(a–d).** This mechanism involves the elimination of a molecule of water to form the alkene **3(a–d)**, where the hydrogen eliminated with OH is acid, since it is in position β with OH and in position α with imine function. Subsequently, the elimination was of order 2, and the H and OH were in anti-coplanar position. It is noted that the yield of the reaction depends on the nature of the substituent R (NO_2_, Cl, CH_3_, and H). It is better in the case of an attractor group by the two inductive and mesomeric effects (R = NO_2_), while it is somewhat smaller for a donor group by mesomeric effect and an attractor group by inductive effect (R = Cl). On the other hand, it is less important for an electron-donating group (R = CH_3_).

## 6. Material and Methods

### 6.1. Chemistry

All reagents were used as obtained from commercial sources. Melting points were determined using a Kofler bench apparatus and are uncorrected. Analytical thin-layer chromatography (TLC) was performed on aluminum plates precoated with E-Merck silica gel 60 F_254_ (St. Somaprol, Morocco). Spectra were recorded with the following instruments: ^1^H NMR spectra: Bruker AC-300; ^13^C NMR spectra: Bruker AC-75 (Bruker, France). Multiplicity was denoted by s (singlet), d (doublet), t (triplet), q (quartet), quin (quintet), dd (double doublet), and m (multiplet). TMS was used as an internal reference and deuteriochloroform as a solvent. High-resolution mass spectra were recorded on a Waters/Vion IMS-QTOF: spectrometer, equipped with an electrospray ionization (ESI), source operating in both positive and negative ion modes. Column chromatography was performed using E-Merck silica gel 60 F_254_ (St. Somaprol, Morocco). Reagents and solvents were purified in the usual manner.

### 6.2. Preparation of 2,2,4-Trimethyl-2.3-Dihydro-1H-1,5-Benzodiazepine 1

Compound **1** is obtained in a yield of 85% by condensation of ortho-phenylenediamine with acetone in acetonitrile according to the reported methods in the literature [31].

### 6.3. Preparation of 1,5-Benzodiazepines **3(a–d)**

A mixture of 100 mmol of BZD **1** and 100 mmol of benzaldehyde derivatives in an acidic medium is brought to reflux of ethanol for 6 h. Afterward, the mixture was extracted with dichloromethane, we purified the product using the silica gel, and then it was recrystallized in ethanol.

(E)-2,2-Diméthyl-4-styryl-2,3-dihydro-1*H*-1,5-benzodiazepine **3a**

Yellow solid, yield = 82%. M.P = 132–133 °C (ethanol), ^1^H NMR (300 MHz, CDCl_3_), δ (ppm) = 1.20, 1.20, (2 s, 6H, -C**H_3_**), 2.30–2.43 (m, -C**H_2_**), 3.01 (1 s, 1H, N**H**), 6.60–7.42 (m, 11H, **H**-Ar). ^13^C NMR (75 MHz, CDCl_3_), δ (ppm) = 30.33, 30.64, (2C, -**C**H_3_), 39.72 (1C, -**C**H_2_-), 68.54 (1C, **C**_IV_), 168.83 (1C, N = **C**_IV_), 121.83, 122.17, 126.26, 127.52, 127.71, 128.28, 128.29, 129.07, 129.31, 132.15, 136.71 (11C, **C**H-Ar), 136.21, 137.99, 141.11 (3C, Ar-**C**_IV_). HRMS of [M + H]^+^ *m*/*z*: calculated for C_19_H_20_N_2_, 277.1626, found: 277.1704.

(E)-2,2-Dimethyl-4-(4-methylstyryl)-2,3-dihydro-1*H*-1,5-benzodiazepine **3b**

Oil product, yield = 80%. ^1^H NMR (300 MHz, CDCl_3_), δ (ppm) = 1.26, 1.26, 2.26 (3 s, 9H, -C**H_3_**), 2.35–2.49 (m, -C**H_2_)**, 3.07 (1 s, 1H, N**H**), 6.64–7.40 (m, 10H, **H**-Ar). ^13^C NMR (75 MHz, CDCl_3_), δ (ppm) = 21.44, 30.35, 30.62, (3C, -**C**H_3_), 39.70 (1C, -**C**H_2_-), 68.46 (1C, **C**_IV_), 168.96 (1C, N = **C**_IV_), 121.84, 122.16, 126.16, 127.66, 128.44, 129.18, 129.47, 129.69, 133.50, 136.79 (10C, **C**H-Ar), 131.19, 138.01, 139.25, 141.17 (4C, Ar-**C**_IV_). HRMS of [M + H]^+^ *m*/*z*: calculated for C_20_H_22_N_2_, 291.1783, found: 291.1861.

(E)-4-(4-Chlorostyryl)-2,2-diméthyl-2,3-dihydro-1*H*-1,5-benzodiazepine **3c**

Oil product, yield = 83%. ^1^H NMR (300 MHz, CDCl_3_), δ (ppm) = 1.17, 1.17 (2 s, 6H, -C**H_3_**), 2.27–2.40 (m, -C**H_2_**), 3.33 (1 s, 1H, N**H**), 6.58–7.27 (m, 10H, **H**-Ar). ^13^C NMR (75 MHz, CDCl_3_), δ (ppm) = 30.30, 30.30, (2C, -**C**H_3_), 39.89 (1C, -**C**H_2_-), 68.26 (1C, **C**_IV_), 168.37 (1C, N = **C**_IV_), 121.78, 122.00, 126.45, 127.85, 128.37, 128.79, 128.85, 129.39, 132.62, 135.11 (10C, **C**H-Ar), 129.49, 130.89, 138.12, 140.67 (4C, Ar-**C**_IV_). HRMS of [M + H]^+^ *m*/*z*: calculated for C_19_H_19_ClN_2_, 311.1237, found: 311.1315.

(E)-2,2-Dimethyl-4-(4-nitrostyryl)-2,3-dihydro-1*H*-1,5-benzodiazepine **3d**

Red solid, yield = 85%. M.P = 173–175 °C (ethanol), ^1^H NMR (300 MHz, CDCl_3_), δ (ppm) = 1.23, 1.24, (2 s, 6H, -C**H_3_**), 2.35–2.49 (m, -C**H_2_**), 3.14 (1 s, 1H, N**H**), 6.62–8.06 (m, 10H, **H**-Ar). ^13^C NMR (75 MHz, CDCl_3_), δ (ppm) = 30.32, 30.64, (2C, -**C**H_3_), 40.27 (1C, -**C**H_2_-), 68.03 (1C, **C**_IV_), 167.52 (1C, N = **C**_IV_), 121.62, 121.90, 123.46, 124.14, 126.95, 127.85, 128.31, 133.44, 133.90, 136.24 (10C, **C**H-Ar), 138.08, 140.11, 142.63, 147.48 (4C, Ar-**C**_IV_). HRMS of [M + H]^+^
*m*/*z*: calculated for C_19_H_19_N_3_O_2_, 322.1477, found: 322.1555.

### 6.4. Crystallographic Data Collection, Structure Solution, and Refinement

Single red prism-shaped crystals of **3d** were used as supplied. A suitable crystal with dimensions 0.19 × 0.17 × 0.13 mm^3^ was selected and mounted on a MiTeGen holder in oil on a SuperNova, Dual, Cu at home/near, HyPix diffractometer (Oxford, UK). The crystal was kept at a steady T = 150.00 (10) K during data collection. The structure was solved with the ShelXT 2018/2 [39] solution program using dual methods and by using Olex2 1.5 [39] as the graphical interface. The model was refined with ShelXL 2018/3 using full matrix least-squares minimization on F^2^. The unit cell was refined using CrysAlisPro 1.171.42.49 [39] on 12,527 reflections, 53% of the observed reflections. Data reduction, scaling, and absorption corrections were performed using CrysAlisPro 1.171.42.49. The final completeness is 100.00% out to 67.075° in Θ. A multi-scan absorption correction was performed using CrysAlisPro 1.171.42.49 (Rigaku Oxford Diffraction, 2022). Empirical absorption correction using spherical harmonics was implemented in the SCALE3 ABSPACK scaling algorithm. The absorption coefficient µ of this material is 0.664 mm^−1^ at this wavelength (*λ* = 1.54184 Å), and the minimum and maximum transmissions are 0.921 and 1.000, respectively.

### 6.5. Computational Method

The geometry of the molecules was optimized by the density functional theory (DFT) method with the Becke-3-Lee-Yang-Parr (B3LYP) functional [40,41] and the 6-31G(d) basis set. To verify the nature of the optimized structures, vibrational frequencies were calculated at the same computational levels without any symmetry restrictions. Localized transition states (TS) are characterized by a single imaginary frequency. An IRC (Intrinsic Reaction Coordinate) calculation [42,43] was performed on the TS to identify the associated local minima on the total potential energy surface (PES). All calculations were performed with the Gaussian 09 program [44].

## 7. Conclusions

This study presents a novel method for synthesizing α,β-unsaturated imine-benzodiazepine derivatives using ethanol as both a solvent and a catalyst. The condensation reaction is highly chemo- and stereoselective, which is the desired product, a satisfactory yield. A simple workup method is sufficient to separate the products, and the chromatographic separation is necessary. The inclusion of electron-drawing substituents into aldehyde’s aromatic ring has a substantial impact on its reactivity. A comprehensive mechanistic examination was also undertaken by DFT methodologies assisted with B3LYP. The reaction pathway was developed based on the obtained energetics, indicating that ethanol is the catalytic agent for the condensation reaction. Furthermore, the results derived from the computation of transition states and the condensation reaction pathways are compatible with experimental data, predicting the creation of thermodynamically and kinetically advantageous products.

## Data Availability

Data is contained within the article or Appendix A.

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
