# Peer review of "A Stereoselective Synthesis of a Novel α,β-Unsaturated Imine-Benzodiazepine through Condensation Reaction, Crystal Structure, and DFT Calculations"

_molecules, 2024, doi:10.3390/molecules29184323_

Round 1
Reviewer 1 Report (Previous Reviewer 3)
Comments and Suggestions for Authors
This manuscript was re-submission with some corrections after being rejected before. Now, after reading the authors’ response toward my comments, I have reviewed this manuscript once again. The reason for rejection of the preceding manuscript was that the research content was not worthy of publication as a paper. Therefore, as long as the logical structure of the paper remains unchanged, my conclusion was not change even if the authors added some explanations or made minor revisions to Table and compound numbers.
I made a structural formula search of the product, and I knew that the compound 2a was a known compound. Moreover, the author also knew that it was a known compound and cited reference 20. I was not noticed this paper at that time. But, in any case, it was self-evident that the reaction proceeded via Path A in Scheme 1 without question. As repeating my previous comment, it would be very interesting if the benzene substituent at the p-position would change and a different reaction would proceed, but in the case of the authors results, the same reaction proceeded even employing p-substituted benzaldehydes, and it would not be worth publishing as a paper.
Next, I tried to find importance in reaction mechanism calculation section whether some part of this manuscript remains still worth appearing in the journal. I looked closely at the energy profile in Figure 5, my impression was that the reaction was not represented correctly. Since structures depicted in Figure 4 were too small, there might be some my misunderstanding for an interpretation of the detailed structure.
Ts2 should be exist two types of transitions, Ts2 (R) and Ts2(S).
Is the structure of In2(S) the optimized geometry? In2(S) did not exist hydrogen bond between O-H and the imine N atom. Isomer E could be formed from Ts3’(R) as I pointed out before. As a result, the configurations of (E) and (Z) were determined regardless of the structures of Ts3 or In2. When the energy difference between more stable In2(S) and Ts(S) was compared that between In2(R) and Isomer-E forming Ts3’(R), how the calculation result change?
Furthermore, the structure of In2(R) in Figure 4 and In2(R) in Figure 5 are different.
After all, Figure 5 did not show appropriate both reaction intermediates and reaction paths, and I wonder whether the optimal calculations were performed. The kinetic study part also would not be worth publishing.
Author Response
As we mentioned, compound 3a has already been reported by F. Obama et al, and we have cited their article in the text (reference [20]). Our work has mainly focused on the synthesis of substituted 2b-d derivatives, which, to our knowledge, are novel compounds. At the same time, we have re-synthesized the 3a product, with the specific aim of using it as a reference to compare our results. We apologize for any confusion this may have caused.
Reviewer 2 Report (Previous Reviewer 2)
Comments and Suggestions for Authors
The comments were included in the attachment file.

Author Response
Comment 1: In Fig. 5, the expression of stereoscopic configuration was not modified correctly. The correct structures were listed as follow.
Response: Thank you for pointing out this typos error, We agree with your comment. Therefore, we have modified the representation of the signaled structures. Please Check the manuscript. (page 10, Fig.5)
Comment 2: In Fig. 5, TS3 (S) and TS3 (R) show the transition states with six-membered ring, but the beta hydrogen of the hydroxyl group is not active hydrogen, and it cannot form a hydrogen bond with the oxygen atom of ethanol. The elimination reaction is more reasonable when the β hydrogen and hydroxyl groups are in trans coplane. These questions should be discussed in the reaction mechanism.
Response: Thank you for pointing this out. We agree with your comment, and as you noted, the elimination mechanism is indeed the most reasonable. Therefore, we have added the following sentence in the paper as you recommended.
Modified text: This mechanism involves the elimination of a molecule of water to form the alkene 3(a-d), where, the hydrogen eliminated with OH is acid, since it is in position b with OH and in position a with imine function. Subsequently, the elimination was of order 2 as well as the H and OH were in anti-coplanar position. (page 10, paragraph 2, line 2)
Reviewer 3 Report (Previous Reviewer 1)
Comments and Suggestions for Authors
Thanks for addressing all my questions.
Author Response
No comments. The comments suggested in the first version have been taken into consideration and already corrected
Round 2
Reviewer 1 Report (Previous Reviewer 3)
Comments and Suggestions for Authors
This manuscript is a revised manuscript with information added by the authors that I previously reviewed and judged inappropriate for publication. My previous conclusion was that this manuscript lacked novelty and the reaction mechanism calculations were inaccurate. Therefore, addition of some description will not change my judgment.
In particular, the authors emphasize the synthesis of new derivatives with different substituents, but unless the physical properties of the compounds obtained are unique, I do not think that the synthesis is generally novel.
This manuscript is a resubmission of an earlier submission. The following is a list of the peer review reports and author responses from that submission.
Round 1
Reviewer 1 Report
Comments and Suggestions for Authors
This paper presents stereoselective formation of unsaturated imine benzodiaziepine complemented by DFT calculations. There are some points need to be addressed and elaborated from the computational standpoint. 1) In Table 2, relative energies for 5 and 6 are more negative pointing out more stable isomers, however, the absolute energies are the other way around. I think the representation is not clear and the definition of delta-E must be given explicitly in the main text. 2) The authors claim that C7 carbon surrounded by heteroatoms is less stable for the reaction to take place, however, they don't explain and rationalize it. 3)In Figure 4, there are two different Ts1 structures with the same labelling. The authors should revisit these structures and report the corresponding one or explain the difference and rename them accordingly.
Reviewer 2 Report
Comments and Suggestions for Authors
The paper reported A Stereoselective Synthesis of a Novel α,β-Unsaturated ImineBenzodiazepine through Condensation Reaction, Crystal Structure, and DFT Calculations. The whole manuscript is smooth and the result is of interest. I recommend the publication of this article after minor revision.
1. The paper, J Struct Chem 64, 2326–2342 (2023), should be cited. In the paper, two similar compounds have been reported.
2. In Fig. 5, the expression of stereoscopic configuration is not accurate (In2 (R), In2 (S), Ts3 (R), Ts3 (S), Isomer Z, and Isomer E). The small heads of the wedges should point towards the chiral carbon atoms. The configuration of the double bonds are not well represented by the wedges. They should be corrected.
3. In Fig. 5, TS3 (S) and TS3 (R) show the transition states with six-membered ring, but the beta hydrogen of the hydroxyl group is not active hydrogen, and it cannot form a hydrogen bond with the oxygen atom of ethanol. The elimination reaction is more reasonable when the β hydrogen and hydroxyl groups are in trans coplane.
Reviewer 3 Report
Comments and Suggestions for Authors
This manuscript is described concerning the synthesis of styryl substituted benzodiazepines by the condensation reaction of a benzodiazepine with benzaldehydes. The authors also presented X-ray crystal structural and DFT calculation study. After reading this manuscript, I thought that this manuscript satisfied neither the readers’ interest nor the value of science. As a result, I concluded that this manuscript will not appear to this journal. I will show the reasons as follows:
The authors stated in the tittle that a novel benzodiazepine was synthesized. Although 2b-d which have substituents on the benzene ring seemed new compounds, unsubstituted 3a was a known compound (Heteroatom Chem., 2023, 3390122). The difference of the condition was whether the synthesis was carried out in ethanol or in methanol. And it is well-known for organic chemists that the same reaction would occur even if substitution groups on the benzene ring would be changed. Moreover, a main molecular structure had already established with X-ray crystal structural analysis in the paper.
The authors also stated that this reaction proceeded stereoselectivity. However, the aldol condensation reaction between an active methyl group compound and benzaldehyde usually produced trans-form formation due to steric hinderance.
The authors carried out various DFT calculations. It is obvious that BZD 1 operates as a nucleophile and benzaldehyde an electrophile. The difference of reactivity according with p-substituents on the benzene ring of benzaldehyde is also obvious: it is of no use discussing with calculation results.
In the energy profile of product formation (Figure 5), intermediate for transaction forms are not only two types (Ts3 (R) and Ts3 (S)). Four types of transaction forms are plausible as shown in the attached file.
In conclusion section, the authors stated, “A simple workup method is sufficient to separate the product, and the chromatographic separation is unnecessary.” However, in the experimental session, purification with silica gel was required. Moreover, product 3a was isolated as oily product (vs. mp 132-133 oC in the literature), and some impurity was contaminated in 1H-NMR spectra: it showed that the product had low purity.
In addition, 2,2,3-trimethylbenzodiazepine should be 2,2,4-trimethylbenzodiazepine in Abstract and Material and Method section. Table 4 was missing.
